# *NSDHL* Frameshift Deletion in a Mixed Breed Dog with Progressive Epidermal Nevi

**DOI:** 10.3390/genes11111297

**Published:** 2020-10-30

**Authors:** Matthias Christen, Michaela Austel, Frane Banovic, Vidhya Jagannathan, Tosso Leeb

**Affiliations:** 1Institute of Genetics, Vetsuisse Faculty, University of Bern, 3001 Bern, Switzerland; matthias.christen@vetsuisse.unibe.ch (M.C.); vidhya.jagannathan@vetsuisse.unibe.ch (V.J.); 2Dermfocus, University of Bern, 3001 Bern, Switzerland; 3Department of Small Animal Medicine and Surgery, College of Veterinary Medicine, University of Georgia, Athens, GA 30602, USA; maustel@uga.edu (M.A.); fbanovic@uga.edu (F.B.)

**Keywords:** *Canis lupus familiaris*, animal model, genodermatosis, dermatology, skin, CHILD syndrome, ILVEN, epidermal nevus, precision medicine

## Abstract

Loss-of-function variants in the *NSDHL* gene have been associated with epidermal nevi in humans with congenital hemidysplasia, ichthyosiform nevi, and limb defects (CHILD) syndrome and in companion animals. The *NSDHL* gene codes for the NAD(P)-dependent steroid dehydrogenase-like protein, which is involved in cholesterol biosynthesis. In this study, a female Chihuahua cross with a clinical and histological phenotype consistent with progressive epidermal nevi is presented. All exons of the *NSDHL* candidate gene were amplified by PCR and analyzed by Sanger sequencing. A heterozygous frameshift variant, c.718_722delGAACA, was identified in the affected dog. In lesional skin, the vast majority of *NSDHL* transcripts lacked the five deleted bases. The variant is predicted to produce a premature stop codon truncating 34% of the encoded protein, p.Glu240Profs*17. The mutant allele was absent from 22 additionally genotyped Chihuahuas, as well as from 647 control dogs of diverse breeds and eight wolves. The available experimental data together with current knowledge about *NSDHL* variants and their functional impact in humans, dogs, and other species prompted us to classify this variant as pathogenic according to the ACMG guidelines that were previously established for human sequence variants. Therefore, we propose the c.718_722delGAACA variant as causative variant for the observed skin lesions in this dog.

## 1. Introduction

Congenital hemidysplasia, ichthyosiform nevi, and limb defects (CHILD) is a well-described syndrome in humans (OMIM #308050) [1,2,3,4,5]. CHILD patients typically have unilateral limb defects together with the characteristic skin lesions termed as CHILD nevi [2]. The severity of the limb defects ranges from aplasia of whole limbs to dystrophy of single finger- or toenails [1,6]. CHILD syndrome is an X-linked semidominant trait. Heterozygous females show the CHILD phenotype, while embryonic lethality is observed in hemizygous males. Due to the random X-chromosome inactivation in heterozygous females, the skin lesions often follow a characteristic pattern, the so-called Blaschko’s lines [7].

To date, a wide range of variants in the *NSDHL* gene have been described as suspected causative variants for the pathogenesis of CHILD syndrome in humans [3,8]. In mice, variants in this gene are responsible for the bare patches (*Bpa*) and striated (*Str*) phenotypes [9]. *Bpa* and *Str* mice do not show limb defects, but their skin phenotype closely resembles CHILD nevi [9]. In veterinary medicine, congenital epidermal nevi without limb defects and candidate causative variants in the *NSDHL* gene have been described in two Labrador Retrievers [10], one purebred Chihuahua [11], and one cat [12] (OMIA 002117-9615, 002117-9685).

The *NSDHL* gene encodes the NAD(P)-dependent steroid dehydrogenase-like protein, which is involved in cholesterol biosynthesis and localized on ER membranes as well as on the surface of lipid droplets [8,13]. A deficiency of this enzyme leads to accumulation of metabolic products from the cholesterol biosynthetic pathway [14,15]. The combination of the toxic effect of those sterol precursors together with decreased cholesterol synthesis and secretion into the stratum corneum is ultimately responsible for the observed skin lesions in CHILD syndrome in humans [16,17].

The currently used therapy for the skin-associated changes in humans and companion animals is aimed at the incorrect cholesterol synthesis. Good results were achieved with topical application of early inhibitors of the cholesterol synthesis pathway in combination with cholesterol-ointments [11,18,19].

In a continuation of our earlier studies [10,11,12], the present study aimed to characterize the clinical and histopathological phenotype in a female Chihuahua cross with congenital progressive epidermal nevi and to identify the likely causative genetic variant.

## 2. Materials and Methods 

### 2.1. Ethics Statement

All animal experiments were performed according to local regulations. The dog in this study was privately owned and was examined with the consent of the owner. The "Cantonal Committee for Animal Experiments" approved the collection of blood samples from control dogs (Canton of Bern; permit 71/19).

### 2.2. Animal Selection

A female spayed Chihuahua cross was investigated. The clinical and dermatological examinations were done by two board certified veterinary dermatologists (M.A. and F.B.). An EDTA blood sample was collected for genomic DNA isolation. Routine histopathological examination of skin biopsies was performed before and after euthanasia of the animal.

Additionally, 22 blood samples of purebred Chihuahuas, which had been donated to the Vetsuisse Biobank were used. They represented unrelated controls without skin lesions. The same 22 dogs had been used as controls in our earlier publication [11]. Furthermore, 655 publicly available canine genome sequences [20] were analyzed as control dataset (Appendix A); 594 of these had been used in our earlier publication [11]. The skin phenotypes of these 655 control dogs were not consistently documented. As the investigated phenotype is rare and at the same time very striking and obvious, we assumed that these animals were all nonaffected. The available data on their phenotype are summarized in Appendix A.

### 2.3. DNA and RNA Extraction

Genomic DNA was isolated from EDTA blood samples with the Maxwell RSC Whole Blood DNA Kit using a Maxwell RSC instrument (Promega, Dübendorf, Switzerland). Total RNA was extracted from skin biopsies using the RNeasy Mini Kit (Qiagen, Hilden, Germany). The RNA was cleared of genomic DNA contamination using the QuantiTect Reverse Transcription Kit (Qiagen). The same kit was used to synthetize cDNA, as described by the manufacturer.

### 2.4. Gene Analysis

The CanFam3.1 dog reference genome assembly and the NCBI annotation release 105 were used. Numbering within the canine *NSDHL* gene corresponds to the NCBI RefSeq accession numbers XM_014111859.2 (mRNA) and XP_013967334.1 (protein).

### 2.5. PCR and Sanger Sequencing

Primer pairs for the amplification of all ten exons of the *NSDHL* gene were described previously [8] and are given in Appendix A. PCR products for each exon were amplified from genomic DNA using AmpliTaq Gold 360 Master mix (Thermo Fisher Scientific, Reinach, Switzerland). PCR products were visualized using a 5200 Fragment Analyzer instrument (Agilent, Basel, Switzerland), which uses capillary electrophoresis to enable accurate sizing and quantification of nucleic acids. All exons of the *NSDHL* gene were analyzed by direct Sanger sequencing of PCR amplicons. After treatment with exonuclease I and alkaline phosphatase, amplicons were sequenced on an ABI 3730 DNA Analyzer (Thermo Fisher Scientific, Reinach, Switzerland). Sanger sequences were analyzed using the Sequencher 5.1 software (Gene Codes, Ann Arbor, MI, USA). For the RT-PCR on skin cDNA, a forward primer located at the boundary of exons 6 and 7 together with a reverse primer located in exon 10 was used (Appendix A). After an initial denaturation of 10 min at 95 °C, 35 cycles of 30 s at 95 °C, 30 s at 60 °C, and 60 s at 72 °C were performed, followed by a final extension step of 7 min at 72 °C. The RT-PCR products were analyzed on a Fragment Analyzer and sequenced as described above.

## 3. Results

### 3.1. Clinical Examination, Necropsy, and Histopathology

A female spayed Chihuahua mix was examined at 7 and 27 months of age. The dog presented with a history of band- to plaque-like cutaneous lesions affecting both sides of the body and head since adoption at 3 months of age. The lesions were sharply demarcated and characterized by alopecia, hyperpigmentation, verrucous hyperplasia with pronounced enlargement of follicular ostia, tan-colored scaling, brown to black crusting and, in some areas, peripheral erythema along with malodor (Figure 1a). Band-like lesions often followed the lines of Blaschko (Figure 1a,b). Although both sides of the body were affected, there was a striking lesion severity lateralization to the right side. A somewhat sharp linear demarcation of the lesions was noted along the ventral midline (Figure 1a,b). The paw pads of both front limbs exhibited severe hyperkeratosis, fissuring, and moderate tissue swelling.

Over the course from the first to the second examination, the severity and extent of epidermal nevi progressed to involve diffuse areas on head, both inner ear pinnae, neck, and dorsal and lateral trunk (Figure 1c,d). Pronounced lesional lateralization of the epidermal nevi on the right side of the body was still present (Figure 1c,d). Lesion-associated pruritus and right front limb lameness were initially not observed but became pronounced over time.

Multiple general physical examinations over time yielded findings within normal limits with the exception of the skin. Hematology results and biochemistry parameters showed mild anemia, leukocytosis, hypergammaglobulinemia, and hypoalbuminemia. Schirmer tear test results remained within normal limits over time. Multiple skin scrapes were negative for ectoparasites. Numerous skin cytologies at various points in time revealed mild to partially severe, chronic, secondary bacterial infections with mainly coccoid bacteria. A bacterial tissue culture grew methicillin-resistant *Staphylocoocus pseudintermedius* at one point in time.

Histopathological examination of multiple skin biopsies revealed massive parakeratotic hyperkeratosis mixed with compact lamellar orthokeratotic hyperkeratosis along with irregular, severe acanthosis with broad rete ridges extending deep into the superficial dermis (Figure 1e,f). The hyperkeratosis and acanthosis were also present in follicular infundibular epithelium and were associated with follicular distention supported by an underlying proliferative granular cell layer and stratum spinosum. There were small to large clusters of neutrophils within the stratum corneum and multilaminated, pustular crusts enclosing mixed bacteria covered portions of the epidermis. Hair follicles with sebaceous and apocrine glands were incarcerated in the dermis. Moderate numbers of melanocytes and melanin-filled macrophages were present at the dermal-epidermal junction with mild pigmentary incontinence in the superficial dermis. The dermis contained occasional periadnexal infiltrates of lymphocytes, plasma cells, and neutrophils with some dermal edema. When correlated with the description of the linear nature of the patient’s lesions, the histopathological findings were consistent with an epidermal nevus.

The patient’s skin underwent histopathological evaluation of different body sites at three different points in time (at 4, 7, and 27 months of age), which all yielded similar results with variations only in regard to the severity of the observed changes and the extent and nature of secondary infections.

Treatments over time included systemic antibiotics (cephalexin, cefpodoxime, clindamycin, and amoxicillin/clavulanic acid), systemic antifungal (fluconazole), antimicrobial shampoos and sprays (chlorhexidine/miconazole), antiseborrheic shampoo and conditioner (gluconolactone-based), antibiotic ointment (mupirocin 2%), topical steroid (betamethasone dipropionate 0.05%), oral glucocorticoid (prednisone), oral vitamin A, oral isotretinoin, oral antihistamine (diphenhydramine), cryotherapy, and interleukin 31 antibody injections, which all yielded minimal to no improvement of clinical signs. Notable exceptions were the interleukin-31 antibody injections, which were accompanied by a reduction in the patient’s pruritus, and oral prednisone, which correlated with significantly improved right front paw tissue swelling and lameness.

Due to the steady decline in the patient’s overall quality of life, the client elected humane euthanasia at 2 years and 3 months of age. Upon necropsy, no pathologic abnormalities besides the cutaneous lesions and mild, generalized lymphadenopathy were found.

### 3.2. Genetic Analysis

As clinical and histopathological findings resembled previously published companion animals with congenital epidermal nevi [10,11,12], we hypothesized that the phenotype in the affected dog was due to a heterozygous variant in the *NSDHL* gene. Hence, *NSDHL* was investigated as the top functional candidate gene.

Sanger sequencing of all ten exons of the *NSDHL* gene of the examined dog identified a single variant in the coding sequence. The variant is a five base pair deletion in exon 9 of the *NSDHL* gene (Figure 2). The genomic variant designation is NC_006621.3:120,752,486_120,752,490delGAACA. It is a frameshift variant, XM_014111859.2:c.718_722delGAACA.

RT-PCR on RNA derived from a lesional skin biopsy of the affected dog yielded an amplicon of the expected size. Sanger sequencing confirmed that the transcripts were normally spliced. We observed pronounced allelic imbalance in lesional skin of the affected dog. The *NSDHL* transcripts were almost exclusively expressed from the mutant allele and lacked the five deleted nucleotides (XM_014111859.2:r.718_722del; Figure 2b,c,e). The frameshift deletion truncates 122 codons (34%) from the open reading frame of the wild-type *NSDHL* transcript, XP_013967334.1:p.(Glu240Profs*17).

We genotyped 22 unaffected Chihuahuas for the frameshift variant. None of these control dogs carried the mutant allele. The mutant allele was also absent from whole-genome sequence data of 647 control dogs of diverse breeds and eight wolves (Appendix A).

## 4. Discussion

In this study, we identified a heterozygous *NSDHL*: c.718_722delGAACA frameshift variant in a mixed breed dog with a severe form of progressive epidermal nevi. The *NSDHL* gene encodes the enzyme NAD(P) dependent steroid dehydrogenase-like, which is involved in cholesterol biosynthesis and has been associated with human CHILD syndrome [8], as well as canine and feline congenital epidermal nevi [10,11,12].

In this single case investigation, we attempted to apply the American College of Medical Genetics and Genomics (ACMG) approved guidelines for the interpretation of sequence variants in human genetics to a dog with the *NSDHL*:c.718_722delGAACA variant [21]. Three arguments support the pathogenicity of the *NSDHL*:c.718_722delGAACA variant.

Computational and predictive data provide a very strong argument for pathogenicity (PVS1) as the frameshift variant is predicted to result in a loss-of-function (LOF) of the *NSDHL* gene [21]. *NSDHL* LOF variants in human and animal patients represent an established mechanism of disease [8,9,10].

Population data indicate that the mutant allele was only present in a single affected dog, while it was absent from more than 600 control dogs. We rate this as a moderate evidence for pathogenicity (PM2) [21].

Other data finally provide additional supporting evidence for pathogenicity (PP4) [21]. The clinical and histopathological skin phenotype is highly specific and *NSDHL* is the only known candidate gene for such a phenotype. Furthermore, the distribution of the skin lesions along Blaschko’s lines and the female sex of the patient strongly point to an X-chromosomal causative variant.

According to the established ACMG criteria, the combinations of one very strong, one moderate, and one supporting type of evidence is sufficient to classify the *NSDHL*: c.718_722delGAACA variant as pathogenic.

To the best of our knowledge, the found variant is the third disease-causing *NSDHL* variant identified in dogs. Previous studies reported a 14 kb deletion spanning the last three exons of *NSDHL* gene [10] and a single missense variant p.Gly234Arg [11] in dogs with congenital epidermal nevi.

## 5. Conclusions

We identified a frameshift variant, *NSDHL*: c.718_722delGAACA as the probable causative variant for progressive epidermal nevi in a single mixed breed dog. Established guidelines for the interpretation of human sequence variants justify the classification of this variant as pathogenic.

## Figures and Tables

**Figure 1 genes-11-01297-f001:**
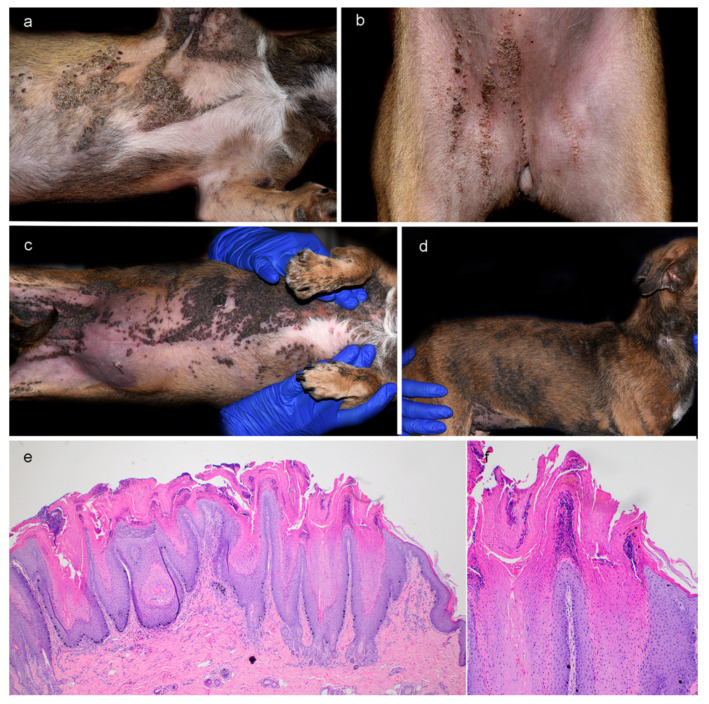
Clinical and histopathological phenotype of progressive epidermal nevi in a Chihuahua mix. (**a**,**b**) Clinical phenotype at 7 months of age. Pronounced lateralization of the skin lesions at the initial presentation on the right axillary area (**a**) and right side of ventral/inguinal area (**b**). (**c**,**d**) Clinical phenotype at 27 months of age. Progression of the severity and extent of epidermal nevi to involve diffuse areas in the axillae, ventral abdomen, and inguinal region (**c**) as well as head, both inner ear pinnae, neck, and dorsal and lateral trunk (**d**). (**e**) Histopathology of a skin biopsy taken at 27 months of age from the ventral neck showing severe parakeratotic hyperkeratosis and acanthosis with broad rete ridges extending deep into the superficial dermis. Follicular infundibula are similarly affected by acanthosis and hyperkeratosis and appear distended. (**f**) Details of the histopathology at higher magnification.

**Figure 2 genes-11-01297-f002:**
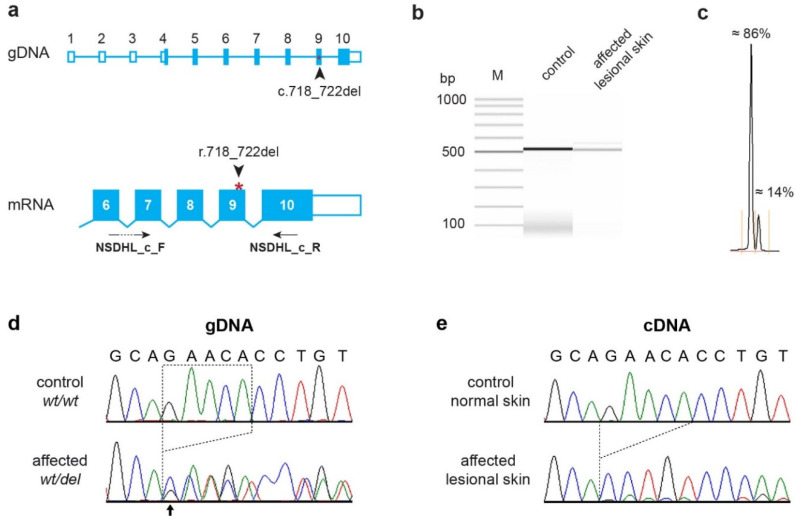
Details of the *NSDHL*:c.718_722delGAACA variant. (**a**) Genomic organization of the canine *NSDHL* gene with 10 annotated exons. The position of the variant on the genomic DNA and the mRNA level are indicated. The position of the primers used for the RT-PCR experiment are also indicated. (**b**) RT-PCR amplification products obtained from cDNA from skin of a healthy control and lesional skin from the affected dog. The samples were analyzed on a FragmentAnalyzer capillary gel electrophoresis instrument. The predominant band in the affected dog corresponds to a correctly spliced transcript lacking 5 nucleotides of coding sequence (r.718_722del). The minor band that migrated slower than the main product most likely represents heteroduplex molecules consisting of one strand containing the 5-nucleotide deletion annealed to a wildtype strand. (**c**) Relative quantification of these two bands. Please note that the ratio of these end-point RT-PCR amplicons should only be seen as a semiquantitative proxy for the true ratio of transcripts. (**d**) Sanger electropherograms derived from genomic PCR products of a control dog and the affected dog. The five bases that are deleted in the affected animal are indicated with dotted lines. The arrow indicates the beginning of overlapping signals in the electropherogram of the affected dog. The intensities of the overlapping signals correspond well to the expected 50:50 allelic ratio for a heterozygous animal. (**e**) Analysis of mRNA. RT-PCR amplicons from the skin of a normal control dog and lesional skin of the affected dog were sequenced. The vast majority of *NSDHL* transcripts of the affected dog lacked 5 nucleotides corresponding to positions 718-722 in the coding sequence. Note the difference in relative signal intensity of the two alleles in the cDNA amplicon compared to the genomic sequence in the affected dog.

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
