# Peer review of "NSDHL Frameshift Deletion in a Mixed Breed Dog with Progressive Epidermal Nevi"

_genes, 2020, doi:10.3390/genes11111297_

Round 1

Reviewer 1 Report

The manuscript NSDHL frameshift deletion in a mixed breed dog with progressive epidermal nevi by Christen et al., is an excellent example of how to transparently report the presumption of causality of a sequence variant discovered by gene sequencing. I particularly appreciate the step by step exposition of lines of evidence of causality in the discussion and the reference 21. Beyond several minor edits, I would like the authors to comment on 3 things. 1) If the dog was seen first at 4 months of age, there should have been an opportunity to examine the dam for similar lesions or a hypomorphic phenotype. Was she examined or reported? 2) Many papers like this one suffer from lack of or disinclination of the investigating laboratory to report effects of a supposed causative variant on mRNA or protein expression. Dermatology cases have the advantage that the gene expressing tissue is on the surface, easily accessible for biopsy and rapid freezing of samples from which to harvest RNA and or protein. If NSDHL was such a strong candidate gene from the beginning of the investigation at Dermfocus, why were not appropriate samples collected? 3) It would further strengthen the paper if the authors would comment on whether the variant is de novo in the patient or illustrates a lack of penetrance in the dam. 

Minor edits:

Line 42, please delete the full title of the disease and rely on the acronym.

Line 85, please remove the 's' on the end of PCR.

Lines 93-102, please indicate at what age this clinical description applies. Later in the manuscript, (line 129) there is reference to histological evaluation at 4 months of age. It seems unlikely this could happen in the absence of a physical examination. Please reconcile.

Line 114, the Latinized names of bacteria stating genus and species must be in italic font.

Line 120, Place a comma after the word "corneum' to join two independent clauses.

Lines 132-140, This sentence is appreciated, but it is over long. Please place a period after the word 'signs' on line 138. Start a new sentence with Notable exceptions ...., and place commas after  'injections,' 'pruritus,' and 'predisone,'. On line 140chnage teh word improved to 'decreased'.

References 6 and 19 need some capital letters removed in the titles to keep the format consistent.

Thank you for allowing me to review this paper.

Author Response

(1)

If the dog was seen first at 4 months of age, there should have been an opportunity to examine the dam for similar lesions or a hypomorphic phenotype. Was she examined or reported?

Response: The patient was left to a shelter at the age of 3-4 months and then adopted. The veterinarian at the shelter examined the dog and performed a skin biopsy. We saw the dog first time at age of 7 months and unfortunately we could not trace back the dam. Clients shared the images and examination results with us from the time they adopted the dog.

(2)

Many papers like this one suffer from lack of or disinclination of the investigating laboratory to report effects of a supposed causative variant on mRNA or protein expression. Dermatology cases have the advantage that the gene expressing tissue is on the surface, easily accessible for biopsy and rapid freezing of samples from which to harvest RNA and or protein. If NSDHL was such a strong candidate gene from the beginning of the investigation at Dermfocus, why were not appropriate samples collected?

Response: We performed an additional experiment investigating the consequences at the mRNA level. The predominant NSDHL transcript in a lesional skin biopsy was normally spliced, but lacked the 5 nucleotides of the genomic deletion. The additional results were integrated in the manuscript.

(3)

It would further strengthen the paper if the authors would comment on whether the variant is de novo in the patient or illustrates a lack of penetrance in the dam.

Response: Unfortunately, as the affected dog was adopted from a shelter, we were not able to trace back the parents of this dog.

(4)

Line 42, please delete the full title of the disease and rely on the acronym.

Response: Revised accordingly.

(5)

Line 85, please remove the 's' on the end of PCR.

Response: Revised accordingly.

(6)

Lines 93-102, please indicate at what age this clinical description applies. Later in the manuscript, (line 129) there is reference to histological evaluation at 4 months of age. It seems unlikely this could happen in the absence of a physical examination. Please reconcile.

Response: Please see our response to comment no. 1. We did not see the dog until it was 7 months old. The earlier clinical and histopathological reports were obtained while the dog was living in the shelter and given to us by the dog’s owner.

(7)

Line 114, the Latinized names of bacteria stating genus and species must be in italic font.

Response: Revised accordingly.

(8)

Line 120, Place a comma after the word "corneum' to join two independent clauses.

Response: Revised accordingly.

(9)

Lines 132-140, This sentence is appreciated, but it is over long. Please place a period after the word 'signs' on line 138. Start a new sentence with Notable exceptions ...., and place commas after 'injections,' 'pruritus,' and 'predisone,'. On line 140 change the word improved to 'decreased'.

Response: Revised accordingly.

(10)

References 6 and 19 need some capital letters removed in the titles to keep the format consistent.

Response: Revised accordingly.

Reviewer 2 Report

In this manuscript, the authors describe a case study of a mixed breed female Chihuahua cross with a severe form of progressive epidermal nevi and present strong evidence that a frame shift mutation in the NSDHL gene is the cause.  The manuscript is very similar to the author’s previous publication (Ref 11) describing a Chihuahua with similar pathology and mutated NSDHL gene.  The manuscript is well written and appropriate for a short communication.  The data presented, together with previously published data, support the conclusion that the mutation is pathogenic in nature.

Major Concerns/Comments:

Did any of the 22 non-affected Chihuahuas, 654 genetically diverse publically available genome sequences, and 8 wolves used for comparison in this study overlap with those used in the authors’ previous publication (ref 11)?

If so, the authors should divulge the nature of the overlap in the methods and discussion.  As the manuscript is currently written, it would appear the genomic comparisons made are new.  However, if the “control” genomes are the same as used in the previous study, then the authors already knew the majority of the “control” samples were wildtype, making it a biased sample population.  Furthermore, the statements made on lines 172-174 would actually represent data previously reported.  While the use of the same controls in both studies does not change the conclusions of the study, it misrepresents the study design. 

Assuming there is overlap in the data sets, the authors need to clearly state the nature of the overlap and present the data as a follow up or continuation of the previous publication, rather than a completely independent study.

Author Response

(1)

Did any of the 22 non-affected Chihuahuas, 654 genetically diverse publically available genome sequences, and 8 wolves used for comparison in this study overlap with those used in the authors’ previous publication (ref 11)?

If so, the authors should divulge the nature of the overlap in the methods and discussion.  As the manuscript is currently written, it would appear the genomic comparisons made are new.  However, if the “control” genomes are the same as used in the previous study, then the authors already knew the majority of the “control” samples were wildtype, making it a biased sample population.  Furthermore, the statements made on lines 172-174 would actually represent data previously reported.  While the use of the same controls in both studies does not change the conclusions of the study, it misrepresents the study design.

Assuming there is overlap in the data sets, the authors need to clearly state the nature of the overlap and present the data as a follow up or continuation of the previous publication, rather than a completely independent study.

Response: We had no intentions to “oversell” this study and apologize, if our presentation of the manuscript was not fully clear. We indeed used overlapping control dogs and control genome sequences between this manuscript and our earlier publication (ref. 11). This is now explicitly stated in chapter 2.2 of the Material and Methods section. We also explicitly stated that the present manuscript represents a continuation of similar earlier studies (line 59).

We do not agree with the reviewer that the reported genotypes were already published (lines 172-174 in the original submission). In Leuthard et al. (ref. 11), we reported the genotypes at NSDHL:c.700G>A. In the present study, we report the genotypes at NSDHL:c.718_722delGAACA. These are different results. Prior to our new study, we only knew that the used controls were “wildtype” at c.700G>A, but not that they were also “wildtype” at the newly discovered deletion as we previously did not specifically analyze their genotypes at the c.718_722 region.

Round 2

Reviewer 2 Report

To the author’s comment regarding the control dog genotypes not being previously reported, the authors are correct.  I mistakenly assumed that when evaluating the 500+ control dogs for the NSDHL:c700G>A variant in the their previous publication, the authors had reported evaluating the entire HSDHL gene for other potential variants.  I see now the authors only reported on the analyses of controls for the exact position of the NSDHL variant identified in the affected dog in the previous study.

In the current study, did the authors actually analyze the entire NSDHL gene for additional variants in the control samples?  If so, they should state this and report these important results.  If not, they should complete this more comprehensive analysis.  Without this full analysis, the authors cannot rule out the presence of alternate, unidentified NSDHL variants located outside the specific NSDHL positions (700 and 718-722) in the unaffected control dogs. This data is critical to support the author's conclusions regarding NSDHL as a candidate gene for this disease and classifying the NSDHL:c.718_722delGAACA variant as pathogenic.

Because the whole genome sequencing data is publicly available for the additional dog and wolf control data sets, variant analysis of the entire canine NSDHL should be possible and relatively easy to complete. Sanger sequencing of all exons for the 22 control chihuahuas would be more time consuming, but important to do considering these are breed matched controls.  I can understand why variant analysis of the entire NSDHL gene may have been overlooked in the previous initial publication, but for this follow up study, the more comprehensive analysis is warranted.

Have there been reports of non-pathogenic variants within the NSDHL gene in either dogs or humans or has every reported variant of this gene been associated with a similar pathology?

Can the authors confirm that the 654 additional control dogs had no clinical signs of ILVEN.  If so, please add that note to the Methods, Animal Section 2.2.

RNA and RT-PCR

Can the authors confirm the methods for RNA extraction?  They start by saying they used the RNeasy kit, but then state the tissue was lysed in TRIZOL.  The RNAeasy kit does not use Trizol, so this is confusing.  There is a modified/hybrid protocol for using the RNeasy spin columns to clean up RNA from Trizol extracted samples, is that what they did?  If that is the case, then the RNA would have been extracted using TRIZOL, then washed and purified using the RNeasy spin columns (assuming that is the protocol they used).  Please clarify and provide a reference or more detailed protocol.

For the RT-PCR, how were the amplicons analyzed?  Where they viewed on an agarose gel or just with sanger sequencing?  The authors mention allelic imbalance and that the mutant allele was almost exclusively expressed.  How was that determined?  Did they perform quantitative PCR?  Please provide a figure for that data.  In Figure 2b, the caption seems to indicate the electropherogram results from Sanger Sequencing show quantitative data, but I am unaware of sanger sequencing being used for quantification.  If this is the case, please provide a reference.

The authors mention an additional PCR amplicon that was 40bps longer.  Can the authors provide an explanation for that amplicon?  If you have multiple amplicons in the PCR product, how were they able to get clean sequencing data?  Furthermore, if there were unexpected amplicons in the PCR reaction, how were they able to accurately quantify the expression of the mutant allele?

Author Response

(1)

In the current study, did the authors actually analyze the entire NSDHL gene for additional variants in the control samples?  If so, they should state this and report these important results.  If not, they should complete this more comprehensive analysis.  Without this full analysis, the authors cannot rule out the presence of alternate, unidentified NSDHL variants located outside the specific NSDHL positions (700 and 718-722) in the unaffected control dogs. This data is critical to support the author's conclusions regarding NSDHL as a candidate gene for this disease and classifying the NSDHL:c.718_722delGAACA variant as pathogenic.

Because the whole genome sequencing data is publicly available for the additional dog and wolf control data sets, variant analysis of the entire canine NSDHL should be possible and relatively easy to complete. Sanger sequencing of all exons for the 22 control chihuahuas would be more time consuming, but important to do considering these are breed matched controls.  I can understand why variant analysis of the entire NSDHL gene may have been overlooked in the previous initial publication, but for this follow up study, the more comprehensive analysis is warranted.

Response: We respectfully disagree with the comment of the reviewer. This manuscript investigates one specific variant within NSDHL, c.718_722del, and provides quite strong evidence that this variant is pathogenic (as outlined in the discussion).

The analysis of the entire NSDHL gene in many control animals is beyond the scope of our manuscript and is an experiment that is of little help to confirm or reject our hypothesis that c.718_722del is pathogenic.

What would be the significance of e.g. a conservative amino acid exchange in a non-conserved part of the NSDHL protein in a control dog? What exactly is the “entire NSDHL” gene? Should this also include the promoter and other regulatory regions?

The previous literature in humans, mice, and our own previous studies in dogs and cats clearly demonstrated that heterozygous NSDHL loss-of-function variants in female animals lead to verrucous epidermal nevi that follow Blaschko lines. The present manuscript identifies one more additional pathogenic variant and provides a comprehensive phenotype characterization for this specific variant in one affected dog.

(2)

Have there been reports of non-pathogenic variants within the NSDHL gene in either dogs or humans or has every reported variant of this gene been associated with a similar pathology?

Response: We are not aware of any publications that systematically investigated “non-pathogenic” variants within the NSDHL gene. This information is rarely published in the scientific literature. The Gnomad database lists 368 variants in the human NSDHL gene without genotype-phenotype correlations. The ClinVar database lists 56 small variants within the NSDHL gene:

23 --> benign or likely benign (e.g. p.Met9Val, p.Leu95Val)

17 --> pathogenic or likely pathogenic (e.g. p.Ala105Val; p.Tyr302*)

  5 --> conflicting interpretations of pathogenicity (e.g. p.Arg199Cys; p.Val243Met)

11 --> uncertain significance

15 of the 17 pathogenic variants lead to CHILD syndrome in humans with comparable skin phenotypes as seen in the affected dog. The remaining 2 human pathogenic variants lead to a completely different clinical phenotype, termed CK syndrome. They probably result in hypomorphic alleles rather than true null alleles as CK syndrome is inherited as an X-linked recessive trait and typically seen in hemizygous males, whereas CHILD syndrome is an X-chromosomal semi-dominant trait with embryonic lethality in hemizygous males.

The bioinformatic analysis of our dog genome sequences identified 310 variants in the canine NSDHL gene (annotated gene and 1000 bp flanking sequence on either side). The automated bioinformatic analysis of illumina whole genome sequence data is likely to yield a small proportion of false positive sequence variants while at the same time it will not accurately identify large structural variants. A validation or further characterization of these 310 variants is beyond the scope of our manuscript.

(3)

Can the authors confirm that the 654 additional control dogs had no clinical signs of ILVEN.  If so, please add that note to the Methods, Animal Section 2.2.

Response: 313 of the 655 (we corrected the number in revision 1) control sequences were derived from dogs that we investigated for other inherited traits and for those dogs we have reliable skin phenotype information. All of these dogs were free of verrucous epidermal nevi. The remaining 343 control sequences were derived from publicly deposited sequence data. For those dogs, we do not have reliable phenotype information regarding the occurrence of verrucous epidermal nevi. We expanded the section describing the control animals and added the available phenotype information to Table S1.

(4)

RNA and RT-PCR

Can the authors confirm the methods for RNA extraction?  They start by saying they used the RNeasy kit, but then state the tissue was lysed in TRIZOL.  The RNAeasy kit does not use Trizol, so this is confusing.  There is a modified/hybrid protocol for using the RNeasy spin columns to clean up RNA from Trizol extracted samples, is that what they did?  If that is the case, then the RNA would have been extracted using TRIZOL, then washed and purified using the RNeasy spin columns (assuming that is the protocol they used).  Please clarify and provide a reference or more detailed protocol.

Response: We apologize for the error. We indeed used the regular protocol of the Qiagen RNeasy kit without any use of Trizol. We removed the sentence with the Trizol step.

(5)

For the RT-PCR, how were the amplicons analyzed?  Where they viewed on an agarose gel or just with sanger sequencing?  The authors mention allelic imbalance and that the mutant allele was almost exclusively expressed.  How was that determined?  Did they perform quantitative PCR?  Please provide a figure for that data.  In Figure 2b, the caption seems to indicate the electropherogram results from Sanger Sequencing show quantitative data, but I am unaware of sanger sequencing being used for quantification.  If this is the case, please provide a reference.

Response: The RT-PCR amplicons were analyzed by fragment size analysis and quantification on a Fragment Analyzer capillary electrophoresis instrument. Subsequently, they were Sanger sequenced to confirm their identity and to get a semi-quantitative measure on the relative abundance of wildtype and mutant transcripts. To the best of our knowledge, the peak area under signals in Sanger electropherograms has been used for many years as a semi-quantitative analysis method to determine e.g. allelic ratios. It does not allow absolutely accurate quantitative measurements, but it is a straightforward and direct method for semiquantitative measurements.

We are not aware of any specific reference, which introduced this widely used methodology. We applied it in many of our own publications and it was always approved during the peer review process of these earlier publications (e.g. Ekenstedt et al. (2014) PLoS Genet 10: e1004635 [Fig. 3]; Dürig et al. (2018) Anim. Genet. 49: 284-290 [Fig. 4]).

We added more details on our experimental procedure to the Materials & Methods section and provide a completely revised and expanded version of figure 2. The new legend to figure 2 should explain the methodological approach also to non-specialists.

Please see also our response to the next comment, which provides additional support for our quantification of the allelic imbalance.

(6)

The authors mention an additional PCR amplicon that was 40bps longer.  Can the authors provide an explanation for that amplicon?  If you have multiple amplicons in the PCR product, how were they able to get clean sequencing data?  Furthermore, if there were unexpected amplicons in the PCR reaction, how were they able to accurately quantify the expression of the mutant allele?

Response: We thank the reviewer for prompting us to think more about this band. We now realized that this minor band represents heteroduplex molecules consisting of one strand of 515 nt mutated cDNA and one strand of 520 nt wildtype cDNA. The “bubble” in these heteroduplex molecules explains the slower migration during capillary electrophoresis. The Sanger sequence data confirmed that there is no molecule that is ~40 bp longer than the 515 bp or 520 bp long specific amplicons.

This gives us now 2 complementary data points for the (semi-)quantitative ratio of the two alleles in the NSDHL transcripts. The ratio of the homoduplex 515 bp band to the 515/520 bp heteroduplex band is approximately 86:14 (new Figure 2c). This would indicate that 7% of the NSDHL mRNA represent the wildtype sequence and 93% are derived from the mutant X‑chromosome with the deletion. (We are aware that the band intensities in an end-point PCR do not accurately reflect the quantitative ratio of the template molecules, but they may serve as a reasonable semi-quantitative proxy. The main band may also include a very small proportion of 520/520 bp wildtype homoduplexes that are not separated from the 515 bp band, but they will not significantly shift the ratio.)

The signal intensities in the Sanger electropherograms are perfectly consistent with a 7 : 93 ratio of the wildtype and mutant alleles in the NSDHL transcripts and thus corroborate the first analysis (new Figure 2e).